# Hybrid Cost Volume for Memory-Efficient Optical Flow

Yang Zhao*
boring_yang@sjtu.edu.cn
Shanghai Jiao Tong University
Shanghai, China

Gangwei Xu*
gwxu@hust.edu.cn
Huazhong University of Science and
Technology
Wuhan, China

Gang Wu†
dr.wugang@sjtu.edu.cn
Shanghai Jiao Tong University
Shanghai, China

## Abstract

Current state-of-the-art flow methods are mostly based on dense all-pairs cost volumes. However, as image resolution increases, the computational and spatial complexity of constructing these cost volumes grows at a quartic rate, making these methods impractical for high-resolution images. In this paper, we propose a novel Hybrid Cost Volume for memory-efficient optical flow, named HCV. To construct HCV, we first propose a Top-k strategy to separate the 4D cost volume into two global 3D cost volumes. These volumes significantly reduce memory usage while retaining a substantial amount of matching information. We further introduce a local 4D cost volume with a local search space to supplement the local information for HCV. Based on HCV, we design a memory-efficient optical flow network, named HCVFlow. Compared to the recurrent flow methods based the all-pairs cost volumes, our HCVFlow significantly reduces memory consumption while ensuring high accuracy. We validate the effectiveness and efficiency of our method on the Sintel and KITTI datasets and real-world 4K (2160 × 3840) resolution images. Extensive experiments show that our HCVFlow has very low memory usage and outperforms other memory-efficient methods in terms of accuracy. The code is publicly available at https://github.com/gangweiX/HCVFlow.

## CCS Concepts

• **Computing methodologies** → **Matching**.

## Keywords

optical flow, cost volume, cost aggregation

**ACM Reference Format:**
Yang Zhao, Gangwei Xu, and Gang Wu. 2024. Hybrid Cost Volume for Memory-Efficient Optical Flow. In *Proceedings of the 32nd ACM International Conference on Multimedia (MM '24), October 28-November 1, 2024, Melbourne, VIC, Australia.* ACM, New York, NY, USA, 10 pages. https://doi.org/10.1145/3664647.3680643

*Equal contribution.
†Corresponding author.

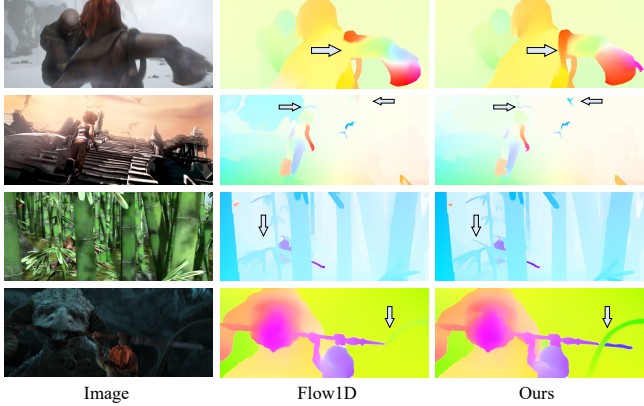

**Figure 1: Qualitative comparisons on the Sintel test set[2]. Compared to the notable memory-efficient method Flow1D[48], our approach achieves more accurate flow estimation in low-texture regions.**

## 1 Introduction

Optical flow, a fundamental aspect of computer vision, aims to estimate the two-dimensional motion of each pixel between two consecutive images. This task serves as a fundamental component providing dense correspondences as valuable clues for downstream applications, including object tracking[26, 39, 41], high-quality video reconstruction[46, 51], and autonomous driving[4, 14, 19]. With the emergence of deep learning, neural network-based methods[9, 15, 18, 31, 33–35, 37, 49] have become mainstream in optical flow algorithms, achieving superior results in accuracy. However, balancing the trade-off between memory consumption and high accuracy remains a challenging endeavor, which limits the application of optical flow algorithms in scenarios involving high-resolution images.

Within the realm of deep learning methods for optical flow estimation, a key module known as the cost volume[9, 18, 34, 35, 37], also referred to as the correlation volume, holds critical importance. This component captures the correlations between pixels across two images, effectively storing a measure of similarity or disparity between them. RAFT[37] represents a significant advancement in the field of optical flow research, constructing a global 4D cost volume by calculating correlations across all pairs of pixels. This cost volume, encompassing comprehensive matching information, has enabled RAFT[37] to achieve remarkable levels of accuracy. However, the approach comes with a drawback: the spatial complexity of building such a cost volume is $O(H \times W \times H \times W)$. With increasing image resolution, the required memory for computation grows quadratically, limiting its application to high-resolution

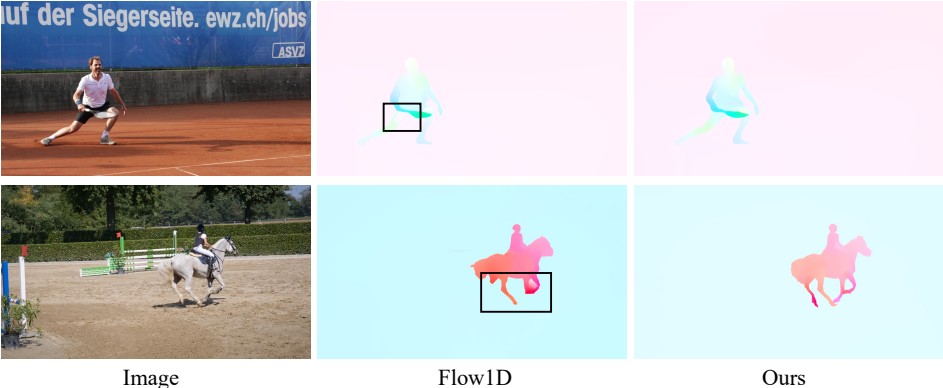

|  |  |  |
|:---:|:---:|:---:|
| Image | Flow1D | Ours |

**Figure 2: Comparisons with Flow1D [48] on high-resolution ($1080 \times 1920$) images from DAVIS[30] dataset. We achieve better results than Flow1D[48] when consuming similar memory.**

images. To mitigate the memory overhead of constructing a full matching cost volume, some researchers have proposed memory-efficient methods for constructing cost volumes[21, 48, 56]. A representative example is Flow1D[48], which constructs two three-dimensional cost volumes along the horizontal and vertical directions, respectively. This approach reduces the overall complexity to $O(H \times W \times (H + W))$. However, in the process of constructing the cost volume, Flow1D[48] utilizes global attention at each position to propagate and aggregate feature information orthogonal to the current row/column. This attention-based aggregation incorporates a substantial amount of noise and is strongly correlated with position, making it struggle to match large motions correctly.

In this paper, we propose a novel Hybrid Cost Volume (HCV) for memory-efficient optical flow estimation. Leveraging this hybrid cost volume, we designed an end-to-end network, named HCVFlow, for optical flow estimation that achieves notable accuracy while requiring reduced memory resources.

The Hybrid Cost Volume (HCV) is constructed in two primary stages: In the first stage, we build two 3D global cost volumes along both horizontal and vertical dimensions by calculating the correlation between the target and reference feature maps. Unlike Flow1D, which aggregates all information into a single value per row or column using attention techniques, our method employs a Top-k strategy. This strategy retains the k positions in each row or column with the highest relevance, ensuring that essential matching information is preserved. Additionally, we introduce a lightweight and efficient separable aggregation module. This module aggregates the 3D cost volumes along both dimensions, capturing more non-local information. This is particularly beneficial for addressing challenges such as occluded areas or large textureless/reflective surfaces. The aggregation module also provides a relatively accurate initial optical flow prediction, laying the groundwork for further optical flow regression. After aggregation, the result is two 3D global cost volumes that encompass a wider array of potential matching scenarios with minimal memory overhead, thereby enhancing the accuracy of subsequent optical flow predictions. In the second stage, a local 4D cost volume is constructed by calculating the correlation within

a local 2D search space. This local 4D cost volume, with a comparatively small search domain, does not significantly increase memory consumption. Importantly, by preserving match information within a localized 2D domain, it complements the global 3D cost volume with critical local details that might be missing otherwise.

By integrating these two global 3D cost volumes with the local 4D cost volume, we achieve the final Hybrid Cost Volume (HCV). This innovative structure effectively balances memory efficiency with the ability to capture detailed motion information, significantly improving both the precision and reliability of optical flow predictions across various challenging scenarios.

The process of constructing two 3D cost volumes with the Top-k strategy incurs an overall complexity of $O(H \times W \times (D + D) \times K)$, where $K$ is substantially smaller than both $H$ (height) and $W$ (width), especially for high-resolution images. We typically set $K$ to 8. D represents the maximum displacement in the horizontal/vertical direction. The complexity for building the local 4D cost volume is $O(H \times W \times (2R + 1)^2)$, where $R$ is the search radius, and $R$ is much smaller than both $H$ and $W$. As a result, the total complexity for constructing the Hybrid Cost Volume (HCV) is maintained at $O(H \times W \times (D + D) \times K)$. The $D$ is smaller than $H$ or $W$ for high-resolution images. In comparison to the $O(H \times W \times H \times W)$ complexity associated with generating a cost volume in RAFT, our methodology significantly reduces memory requirements while capturing the essential matching information effectively.

The experimental results demonstrate that our HCVFlow, constructed using our Hybrid Cost Volume (HCV), achieves remarkable accuracy and exceptionally low memory consumption. The experiments conducted on the KITTI[29] datasets showed that our method outperforms previous memory-efficient methods[21, 48], such as Flow1D[48], by more than 16%. The accuracy of our model are close to those of RAFT[37], yet it only requires one-eighth of the memory used by RAFT. Furthermore, our benchmark tests on the Sintel[2] test dataset have surpassed RAFT, significantly exceeding other memory-efficient methods. Specifically, our method outperforms Flow1D by 26% on Sintel (Final) test dataset.

Overall, our work makes the following key contributions:

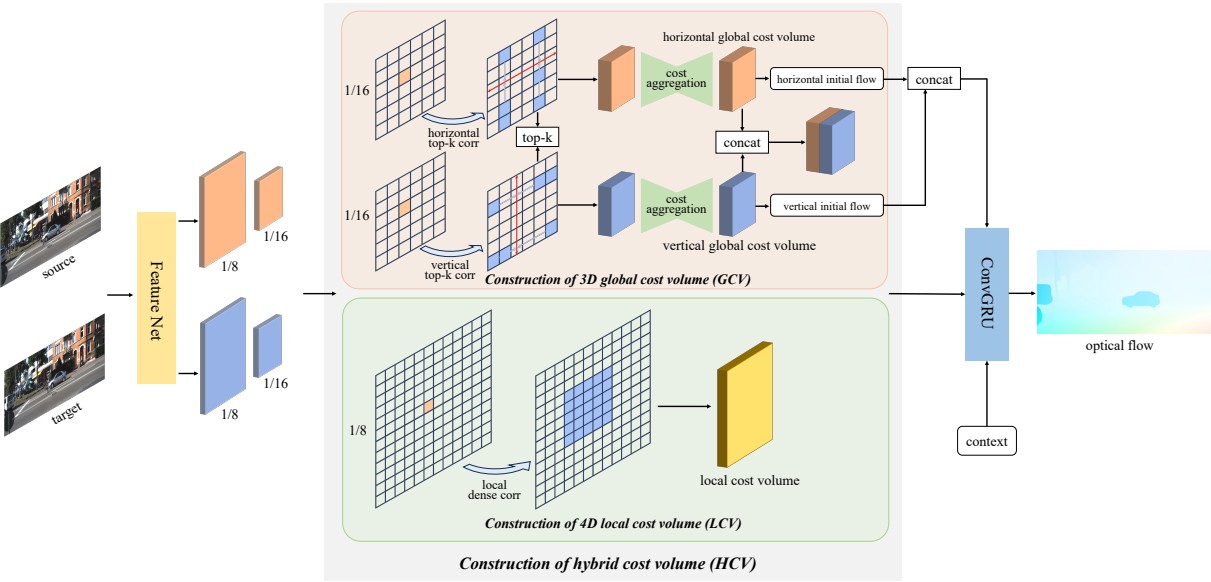

**Figure 3: Overview of our HCVFlow. We obtain feature maps at 1/8 and 1/16 resolutions and construct the Hybrid Cost Volume (HCV) using these feature maps. Specifically, we compute initial cost volumes in both horizontal and vertical directions, followed by obtaining 3D cost volumes through a Top-k strategy. Subsequently, we aggregate these volumes using an aggregation module to obtain the final 3D global cost volume. Additionally, we construct a 4D local cost volume. Finally, we input the the hybrid cost volume and initial flow predictions generated by aggregation module into the ConvGRU module for iterative flow prediction.**

- We develop a memory-efficient technique for constructing cost volumes by implementing a Top-k strategy. This approach allows us to decompose the conventional 4D cost volume into two more manageable 3D cost volumes, significantly reducing memory requirements while preserving the most valuable matching information. Additionally, we designed a lightweight aggregation module that enables these 3D cost volumes to capture more non-local information, enhancing their capacity to account for complex motion scenarios.

- Innovatively, we combine the 3D global and 4D local cost volumes to create the Hybrid Cost Volume (HCV). This novel structure not only minimizes memory consumption but also encodes a rich set of effective matching information capable of handling various motions. The integration of global and local cost volumes addresses the challenges of accurately predicting motion across a wide range of scenarios, making HCV a versatile and efficient solution.

- Leveraging the Hybrid Cost Volume (HCV), we have constructed an end-to-end optical flow prediction network named HCVFlow. Experimental results demonstrate that HCVFlow surpasses representative memory-efficient method Flow1D by 16% in terms of accuracy on KITTI dataset, closely rivaling the performance of RAFT with only one-eighth of RAFT's memory consumption. On the Sintel test dataset, HCVFlow's benchmark results exceed Flow1D by more 26% and also surpass RAFT.

## 2 Related Work

### 2.1 Deep-Flow Method

The research of optical flow has a long history, with traditional methods being explored for optical flow estimation decades ago. Among these, the Horn-Schunck[13] and Lucas-Kanade[24] methods stand out as seminal approaches. However, in recent years, the advent of deep learning has led to a surge of techniques based on this paradigm, which have significantly outperformed traditional methods in terms of accuracy. As a result, deep learning-based optical flow prediction methods[9, 11, 15, 18, 25, 28, 33, 36, 46] has become dominant in the field.

FlowNetS[9] is pioneering work in the end-to-end prediction of optical flow using CNN technology, which follows a straightforward, end-to-end learning approach without any specialized layers or mechanisms specifically for optical flow beyond the standard convolutional layers. FlowNetC[9] introduces a correlation layer to better capture the relationship between two images by learning a similarity measure. FlowNet 2.0[18] uses a stacked architecture that refines optical flow estimates through multiple training schedules to achieve high accuracy. PWC-Net[34] leverages pyramidal processing, warping, and cost volume layers which enables efficient handling of motions at different scales and has shown remarkable performance, especially in scenarios with rapid movement and occlusions.

A recent notable work is RAFT[37], which introduces an all-pairs cost volume. This cost volume stores a wealth of matching information, allowing RAFT to achieve higher accuracy during flow regression with GRU block. RAFT achieves state-of-the-art accuracy in the optical flow, particularly outperforming other methods in

challenging scenarios such as fast-moving objects and occluded areas. Following RAFT's success, several researchers have made improvements and innovations based on its fundamental structure. However, the construction cost of this all-pairs 4D cost volume, which is $O(H \times W \times H \times W)$, results in high memory usage, making these methods challenging to apply to high-resolution images due to the significant GPU memory consumption.

## 2.2 Memory-efficient Method

To facilitate the use of optical flow algorithms on lower-end consumer GPUs and with high-resolution images, recent research[21, 42, 48] has introduced a series of studies focused on developing memory-efficient optical flow networks. SCV[21] adopts a sparse cost volume to replace RAFT's all-pair cost volume, aiming to decrease memory usage. For each position, SCV utilizes a Top-k strategy, keeping only the k most relevant points for subsequent matching. However, in areas that lack distinctive features or are blurry, the inherent ambiguity can lead to a multitude of potential matches. In such cases, the Top-k approach may not encompass the correct match, potentially leading to erroneous motion predictions.

Flow1D innovatively proposed replacing the all-pairs 4D cost volume with two separate 3D cost volumes[6, 7, 12, 32, 43–45] along the horizontal and vertical directions, significantly reducing the network's memory usage. However, in constructing the cost volume, Flow1D utilized the attention mechanism to aggregate information from each row/column into a singular value. This approach led to a loss of critical information in the resulting cost volume, making it susceptible to mismatches, particularly in long-distance, high-speed motions where Flow1D often predicts incorrectly. The limited local information also presents a challenge in achieving precise matching. Consequently, despite its advancements, the overall accuracy and generalization capabilities of Flow1D still fall short when compared to RAFT.

In response to these challenges, particularly the lower accuracy and difficulties encountered in certain scenarios by memory-efficient methods[21, 48], our research introduces a novel hybrid cost volume approach. This method combines the global 3D cost volumes with the local 4D cost volume to adeptly manage a variety of motions. The empirical results from testing our network on datasets such as KITTI[29] and Sintel[2] have shown notable enhancements in both generalization and accuracy over prior memory-efficient solutions.

## 3 Method

### 3.1 Hybrid Cost Volume Construction

The construction of the Hybrid Cost Volume (HCV) consists of the following steps: initial 3D global cost volume construction via Top-k strategy; aggregation of the initial 3D global cost volume; and construction of the 4D local cost volume.

**3D global cost volume construction.** We provide a detailed description of the 3D global cost volume construction along the horizontal direction, while the construction for the vertical is similar.

For the input feature maps $\mathbf{F}'_1 \in \mathbb{R}^{C \times H \times W}$ and $\mathbf{F}'_2 \in \mathbb{R}^{C \times H \times W}$, we compute the correlation between feature points using the dot product operation, $H$ and $W$ represent the height and width of the 16× downsampled image respectively. Specifically, the correlation

$C(u, v, i, j)$ between the pixel at $(u, v)$ in $\mathbf{F}'_1$ and the pixel at $(i, j)$ in $\mathbf{F}'_2$ is formulated as:

$$\mathbf{C}(u, v, i, j) = \frac{\mathbf{F}'_1(u, v) \cdot \mathbf{F}'_2(i, j)}{\sqrt{C}}. \tag{1}$$

The $\cdot$ symbol refers to the dot product operation, and $\frac{1}{\sqrt{C}}$ acts as a normalization factor.

For any given horizontal displacement $d$ ($d \in \{-D, -(D-1), -(D-2), \ldots, 0, \ldots, D-1\}$), we first utilize the following formula to calculate the horizontal correlation $\mathbf{C}_{horizontal}$ between the point $(h, w)$ in $\mathbf{F}'_1$ and all corresponding points in $\mathbf{F}'_2$ at a horizontal displacement of $d$:

$$\mathbf{C}_{horizontal}(h, w, d) = \{\mathbf{C}_1, \mathbf{C}_2, \ldots, \mathbf{C}_H\}, \\ \mathbf{C}_i = \mathbf{C}(h, w, d, i). \tag{2}$$

Then we obtain a dense horizontal cost volume $\mathbf{C}_{horizontal}$. To reduce memory consumption while preserving the majority of valuable matching information, we propose to use a Top-k strategy on $\mathbf{C}_{horizontal}$, retaining only the top $k$ points with the highest correlation at a given horizontal displacement $d$. We have

$$\mathbf{C}_{horizontal}^{topk}(h, w, d) = \text{TopK}\left(\mathbf{C}_{horizontal}(h, w, d)\right), \tag{3}$$

where the notation $\text{TopK}(\cdot)$ denotes the operation of selecting the largest $K$ values from a given list. The spatial complexity of our sparse horizontal cost volume, $\mathbf{C}_{horizontal}^{topk}$, is $O(H \times W \times D \times K)$. $D$ represents the maximum horizontal displacement on the left or right. $K$, with a default value of 8, is significantly smaller than both $H$ and $W$ in high-resolution images. Thus, we ultimately obtain a 3D global cost volume $\mathbf{C}_{horizontal}^{topk}$ with a spatial complexity of $O(H \times W \times D \times K)$.

By applying this construction method to the vertical direction, we can easily obtain the vertical 3D global cost volume, $\mathbf{C}_{vertical}^{topk}$.

**Aggregation of the 3D global cost volume.** By employing the Top-k strategy, we have separately obtained the 3D cost volumes for both horizontal and vertical directions. The structure and properties of these unidirectional cost volumes resemble those found in stereo matching[5, 10, 12, 40, 43–45, 47], inspiring us to adopt aggregation methods commonly used in stereo matching to optimize our initial 3D cost volumes.

Therefore, we design a novel, lightweight aggregation module $\mathbf{R}$ to capture more non-local information, enhancing accuracy in handling complex scenarios such as occlusions and textureless regions. The cost aggregation is expressed as,

$$\mathbf{C}_H = \mathbf{R}(\mathbf{C}_{horizontal}^{topk}), \\ \mathbf{C}_V = \mathbf{R}(\mathbf{C}_{vertical}^{topk}). \tag{4}$$

The $\mathbf{C}_H$ and $\mathbf{C}_V$ represent the aggregated horizontal 3D cost volume and vertical 3D cost volume respectively. To implement cost aggregation, we first employ a sequence of 3D convolutions with batch normalization and ReLU activations to downsample the feature maps while further extracting features. Then, we utilize a 3D transposed convolution layer, which enlarges the spatial dimensions of the feature maps, enriching them with spatial information.

**Table 1: Ablation study. GCV denotes global cost volume, LCV denotes local cost volume. The final method, GCV+LCV, is denoted as HCV.**

| Method | Sintel (train, clean) | | | Sintel (train, final) | | | KITTI (train) | | $448 \times 1024$ | | $1080 \times 1920$ | |
|---|---|---|---|---|---|---|---|---|---|---|---|---|
| | EPE | $s_{0\text{-}40}$ | $s_{40+}$ | EPE | $s_{0\text{-}40}$ | $s_{40+}$ | EPE | F1-all | Memory (G) | Time (ms) | Memory (G) | Time (ms) |
| GCV | 1.70 | 0.87 | 9.43 | 3.29 | 1.54 | 19.60 | 7.46 | 25.56 | 0.32 | 60 | 1.38 | 230 |
| LCV | 1.76 | 0.77 | 10.90 | 3.41 | 1.48 | 21.30 | 6.51 | 20.07 | 0.32 | 65 | 1.39 | 290 |
| GCV+LCV (Ours) | **1.51** | **0.74** | **8.67** | **2.84** | **1.23** | **17.79** | **5.33** | **16.80** | 0.38 | 85 | 1.56 | 340 |

**Table 2: Ablation study. The first column represents the strategies used when constructing the 3D global cost volume, corresponding to retaining only the average value, the maximum value, and the top k values of each row/column correlation (in our experiments, k is set to 8). The strategy used in HCV is the Top-k strategy.**

| Strategy | Sintel (train) | | KITTI (train) | |
|---|---|---|---|---|
| | Clean | Final | EPE | F1-all |
| Mean | 1.55 | 2.88 | 5.63 | 18.26 |
| Max | 1.68 | 2.92 | 6.08 | 18.94 |
| Top-k (k=8) | **1.51** | **2.84** | **5.33** | **16.80** |

By developing these two 3D global cost volumes via Top-k strategy, not only can we capture the vast majority of valuable global match information, but we also substantially decrease the memory usage compared to the 4D global cost volume method employed by RAFT. Furthermore, we leverage the $\mathbf{C}_H$ and $\mathbf{C}_V$ for an initial optical flow estimation $\mathbf{f}_{init}$:

$$\mathbf{f}_{init\_h} = \sum_{d=-D}^{D-1} d \times Softmax(\mathbf{C}_H(d)),$$

$$\mathbf{f}_{init\_v} = \sum_{d=-D}^{D-1} d \times Softmax(\mathbf{C}_V(d)), \tag{5}$$

$$\mathbf{f}_{init} = Concat\{\mathbf{f}_{init\_h}, \mathbf{f}_{init\_v}\}.$$

These initial flow estimation $\mathbf{f}_{init}$ provides a relatively accurate starting point for the subsequent flow regression process.

**4D local cost volume construction.** Our 3D global cost volume introduced before is constructed from 1D horizontal and vertical directions. This method, while beneficial for computational efficiency and memory conservation, may result in the loss of certain matching details. Consequently, this approximation can introduce minor inaccuracies in localized regions, potentially affecting the overall precision of the optical flow predictions. To address this limitation, we propose the construction of a local 4D cost volume. Unlike the 3D global cost volume which is constructed at 1/16 resolution, the 4D local cost volume is constructed at 1/8 resolution.

Initially, the 1/8 resolution source feature map $\mathbf{F}_1$ ($\mathbf{F}_1 \in \mathbb{R}^{C \times 2H \times 2W}$) and target feature map $\mathbf{F}_2$ ($\mathbf{F}_2 \in \mathbb{R}^{C \times 2H \times 2W}$) are aligned and compared within a small local range ($l_r$) to compute a correlation score. This process begins by padding the second feature map to ensure that comparisons can be made across all valid positions, followed by

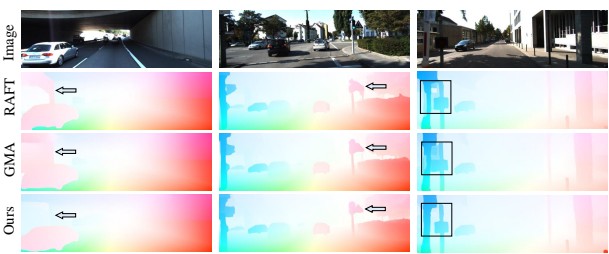

**Figure 4: Qualitative comparisons with accuracy-oriented methods on the KITTI test set[29]. Our novel aggregation module aggregates contextual information to reduce mismatches, thus our method outperforms RAFT and GMA in real-world complex texture-less areas.**

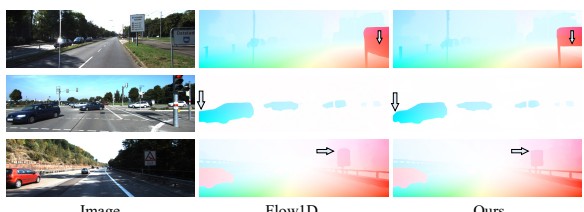

**Figure 5: Qualitative comparisons with memory-efficiency method Flow1D on the KITTI test set[29]. Flow1D fails to accurately predict motion near object edges, while our method can precisely estimate local details.**

an unfolding operation that prepares the feature map for efficient local comparisons. The unfolding radius is $l_r$. The unfolded target feature map is denoted as $\mathbf{F}_2^u \in \mathbb{R}^{C \times 2H \times 2W \times (2l_r+1)^2}$. Then, the 4D local cost volume $\mathbf{C}_L \in \mathbb{R}^{2H \times 2W \times (2l_r+1)^2}$ is constructed by,

$$\mathbf{C}_L(h, w) = \frac{\mathbf{F}_1(h, w) \cdot \mathbf{F}_2^u(h, w)}{\sqrt{C}}. \tag{6}$$

This 4D local cost volume offers a detailed and rich representation of similarity scores across local regions between two feature maps. By integrating this 4D local cost volume, HCV is able to capture more local matching information and enable more precise matching of similar areas between images, thereby reducing the likelihood of mismatches.

## 3.2 HCVFlow Architecture

Following RAFT, we first utilize a feature extraction network to derive feature maps $\mathbf{F}_1$ and $\mathbf{F}_2$ from the original reference and target images, achieving an 8× downsampling. To further reduce the memory footprint for the 4D global cost volume construction, an additional downsampling stage by a factor of 2× is applied to obtain $\mathbf{F}_1'$ and $\mathbf{F}_2'$. Additionally, we incorporate a context network to extract contextual information, which is instrumental for the flow regression. Subsequently, as described in Sec. 3.1, we construct two separate 3D global cost volumes and one 4D local cost volume. By concatenating these volumes, we obtain the core component of HCVFlow, the Hybrid Cost Volume (HCV).

In the flow regression stage, we index correlation features from 3D global cost volumes, $\mathbf{C}_H$ and $\mathbf{C}_V$, and 4D local cost volume $\mathbf{C}_L$. The indexed hybrid correlation features, together with flow features and context features, are all inputted into the flow regression module. Here, we employ a Convolutional Gated Recurrent Unit (ConvGRU[37]) to iteratively update the flow predictions, which leads to the output of the refined optical flow results.

## 3.3 Loss Function

We first compute the L1 loss on the initial optical flow estimation obtained from the aggregation module, defining it as follows:

$$\mathcal{L}_{init} = ||\mathbf{f}_{init} - \mathbf{f}_{gt}||_1, \tag{7}$$

Following RAFT[37], we calculate the L1 loss between all predicted optical flow sequences and the ground truth. Similar to RAFT, we exponentially increase the weights, with $\gamma$ set to 0.8 in our experiments. The loss for the predicted flow is calculated as follows:

$$\mathcal{L}_{iter} = \sum_{i=1}^{N} \gamma^{N-i} ||\mathbf{f}_i - \mathbf{f}_{gt}||_1, \tag{8}$$

Finally, we add these two parts of the loss to obtain the total loss function:

$$\mathcal{L}_{total} = \mathcal{L}_{init} + \mathcal{L}_{iter}. \tag{9}$$

## 4 Experiments

## 4.1 Experimental Setup

**Datasets and evaluation setup.** We conduct experiments on the KITTI[29], Sintel[2], and high-resolution DAVIS[3, 30] datasets to evaluate the effectiveness of our method. We first train our model on the FlyingChairs[9] and FlyingThings3D[27] datasets. Upon completing the training, we conduct extensive experiments on both the KITTI and Sintel datasets to verify the performance and generalization ability of our method. Subsequently, we fine-tune our trained model on a mixed dataset comprising HD1K[22], KITTI, Sintel, and FlyingThings3D and submit it to the KITTI and Sintel websites for benchmark testing. We employ the End-Point Error (EPE) metric to evaluate the model's prediction accuracy on the Sintel dataset and use both EPE and F1-all metrics to evaluate the accuracy on the KITTI dataset. F1-all denotes percentage of outliers for all pixels. Finally, we validate the performance of our method on high-resolution images (1080P and 4K resolutions) using the DAVIS[3, 30] dataset.

**Implementation details.** We implement our HCVFlow using the PyTorch framework, with Adam[23] serving as the optimizer. Our feature network implementation is followed by RAFT[37], but we have added an additional downsampling layer, resulting in a feature map that is downsampled by a factor of 16. Similar to other optical flow methods[32, 37, 48], we trained our model for 100K iterations on the FlyingChairs dataset with a batch size of 12. Then, we trained our model for another 100K iterations on the FlyingThings3D dataset with a batch size of 6. We finally fine-tuned our model on a mixed dataset comprising FlyingThings3D, Sintel, KITTI, and HD1K. For the Sintel evaluation, the fine-tuning was carried out over 100K iterations, and for the KITTI evaluation, it was conducted over 50K iterations. The batch size was set to 6 for fine-tuning. During training, we employed 12 GRU-based iterations. For the evaluation phase, we used 32 GRU-based iterations for Sintel and 24 GRU-based iterations for KITTI, respectively. When building the 3D global cost volume, we utilize a Top-k approach with the k parameter set to 8.

## 4.2 Ablation Study

We carry out ablation studies to confirm the effectiveness and efficiency of HCV's key components. For these studies, models are trained on the FlyingChairs and FlyingThings3D datasets and subsequently evaluated on the Sintel and KITTI training sets. Across all experiments, memory consumption and inference time are measured using 12 GRU-based iterations on our RTX 3090.

We initially verify the effectiveness of the two primary components of HCV: the 3D global cost volume and the 4D local cost volume. As shown in Table 1, the network constructed solely with the global cost volume (GCV) performs well in handling large motions($s_{40+}$), yet shows weak performance for small motions ($s_{0-40}$). Conversely, the network utilizing only the local cost volume (LCV) demonstrates good performance on short-distance movements but struggles with long-distance movements. Our proposed Hybrid Cost Volume (HCV), which concates both GCV and LCV, manages to integrate the advantages of both cost volumes, achieving a synergistic effect where the whole is greater than the sum of its parts. Whether dealing with large motions or small motions, our HCV consistently delivers optimal performance.

Subsequently, we conduct experiments to validate the effectiveness of the Top-k strategy used in constructing the 3D global cost volume. In the construction process, we experiment with three different strategies for retaining the correlation values: the maximum correlation value, the average correlation value, and the Top-k correlation values (with $k = 8$ in our experiments), while keeping all other parameters constant. As shown in Table 2, models constructed using the Top-k strategy outperforms those built with either the maximum or average values across various accuracy metrics on both the Sintel and KITTI datasets. Experimental results demonstrate that our Top-k strategy, which selectively preserves matches with higher correlation, yields more precise estimations than straightforward aggregation approaches like averaging or attention mechanisms, the latter of which tend to incorporate noise.

**Table 3: Comparison with existing representative cost volumes. Memory and are measured for** $448 \times 1024$ **and** $1080 \times 1920$
**resolutions on RTX 3090 GPU, and the GRU-based iteration numbers are 12 for RAFT, Flow1D and our HCVFlow. Bold: Best,**
Underscore: **Second best.**

| Method | Sintel (train) | | KITTI (train) | | Param (M) | Memory (G) | |
|---|---|---|---|---|---|---|---|
| | Clean | Final | EPE | F1-all | | $448 \times 1024$ | $1080 \times 1920$ |
| RAFT[37] | **1.43** | **2.71** | **5.04** | 17.40 | **5.26** | 0.48 | 8.33 |
| FlowNet2[18] | 2.02 | 3.14 | 10.06 | 30.37 | 162.52 | 1.31 | 3.61 |
| PWC-Net[34] | 2.55 | 3.93 | 10.35 | 33.67 | 9.37 | 0.86 | 1.57 |
| Flow1D[48] | 1.98 | 3.27 | 6.69 | 22.95 | 5.73 | **0.34** | **1.42** |
| HCVFlow (Ours) | 1.51 | 2.86 | 5.33 | **16.80** | 6.06 | 0.38 | 1.56 |

**Table 4: Comparisons with memory-efficient methods. Our method demonstrates either the best or the second-best performance**
**across various datasets in terms of accuracy, memory consumption, and inference time. Bold: Best,** Underscore: **Second best.**

| Method | KITTI test | Sintel (test) | | $448 \times 1024$ | | $1080 \times 1920$ | |
|---|---|---|---|---|---|---|---|
| | | Clean | Final | Memory (G) | Time(ms) | Memory (G) | Time(ms) |
| SCV[21] | 6.17 | 1.72 | 3.60 | 0.59 | 280 | 2.66 | 900 |
| DIP[56] | **4.21** | **1.67** | 3.22 | 0.67 | 180 | 2.90 | 620 |
| Flow1D[48] | 6.27 | 2.24 | 3.81 | **0.34** | **52** | **1.42** | **200** |
| HCVFlow (Ours) | 5.54 | 1.69 | **2.81** | 0.38 | 85 | 1.56 | 340 |

**Table 5: Comparisons with accuracy-oriented methods. Our method achieves accuracy levels close to those of accuracy-oriented**
**approaches while significantly reducing memory consumption. At a resolution of 1920x1080, our memory usage is 5 times**
**lower than RAFT's and 7 times lower than SepaFlow's.**

| Method | KITTI test | Sintel (test) | | $448 \times 1024$ | | $1080 \times 1920$ | |
|---|---|---|---|---|---|---|---|
| | | Clean | Final | Memory (G) | Time(ms) | Memory (G) | Time(ms) |
| GMFlow[49] | 9.32 | 1.74 | 2.90 | 1.31 | 115 | 8.30 | 1242 |
| SKFlow[36] | 4.84 | 1.28 | **2.23** | 0.66 | 138 | 11.73 | 634 |
| FlowFormer[15] | 4.87 | **1.18** | 2.36 | 2.74 | 250 | OOM | - |
| RAFT[37] | 5.10 | 1.61 | 2.86 | 0.48 | **64** | 8.33 | **300** |
| GMA[20] | 5.15 | 1.39 | 2.47 | 0.65 | 75 | 11.73 | 387 |
| SepaFlow[54] | **4.64** | 1.50 | 2.67 | 0.65 | 570 | 12.13 | 3948 |
| HCVFlow (Ours) | 5.54 | 1.69 | 2.81 | **0.38** | 85 | **1.56** | 340 |

## 4.3 Comparison with Existing Methods

**Comparison with existing representative cost volumes.** The
core of our approach lies in the construction of a hybrid cost volume.
So we conduct extensive experimental comparisons with other
existing representative cost volume construction methods. We test
the accuracy and memory consumption of RAFT[37], FlowNet2[18],
PWC-Net[34], Flow1D[48], and our HCVFlow on the KITTI and
Sintel datasets on RTX 3090. Our comparison involves models that
employ various methods for constructing cost volumes. Specifically,
RAFT constructs a 4D global cost volume, FlowNet2 generates a
single-scale cost volume, PWC-Net develops a coarse-to-fine cost
volume pyramid, and Flow1D uses an attention mechanism to build
two 3D global cost volumes.

As illustrated in Table 3, our HCVFlow achieves suboptimal End-
Point Error (EPE) on both the Sintel and KITTI datasets, slightly

behind RAFT. However, it surpasses RAFT on the KITTI dataset in
terms of the F1-all metric, achieving the best performance. Our
method outperforms Flow1D by 26.8% and PWC-Net by 50.1%
on the KITTI dataset in terms of the F1-all metric. Moreover, our
method consumes significantly less memory compared to approaches
like RAFT. For images at 1080P resolution, our memory usage is
only one-fifth of RAFT's. Compared to Flow1D, which is also known
for its memory efficiency, our method consumes only slightly more
memory but significantly surpasses Flow1D in terms of accuracy.

**Comparison with memory-efficient methods.** We further com-
pare our method with other memory-efficient optical flow methods.
As shown in Table 4, our HCVFlow exhibits superior accuracy com-
pared to Flow1D[48] and SCV[21]. While we are slightly behind
DIP[56] on some accuracy metrics, our model significantly sur-
passes DIP in terms of memory usage and inference time. When

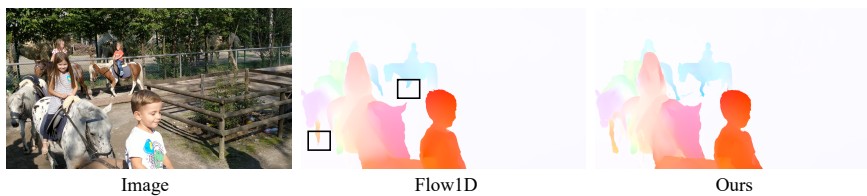

Image               Flow1D               Ours

**Figure 6: Comparisons with Flow1D [48] on high-resolution ($2160 \times 3840$) images from DAVIS dataset.Our approach accurately addresses occluded regions and complex lighting conditions.**

processing 1080P images, our memory consumption is half that of DIP, and our inference time is only half as long.

**Comparisons with accuracy-oriented methods.** We also compare HCVFlow with several accuracy-oriented methods. As shown in Table 5, our accuracy on the KITTI and Sintel test datasets is close to these methods, with our EPE on the Sintel final dataset even surpassing some approaches like RAFT[49] and GMFlow[49]. Moreover, our method exhibits a clear advantage in terms of memory consumption and inference speed. During the inference process for 1080P images, our method's memory consumption is only one-seventh of SKFlow[36]'s and one-eighth of SepaFlow[54]'s. FlowFormer[15] consumes seven times more memory than our method for images of 448*1024 resolution and even fails to process images of 1080*1920 resolution due to OOM (given our RTX 3090's maximum memory capacity is 24G).

### 4.4 Benchmark Results

In Table 6, we showcase the benchmark results of our method on the KITTI and Sintel test datasets, along with comparisons to other methods. Our accuracy surpasses most methods, including PWC-Net+[35], SCV[21], Flow1D[48], and MaskFlowNet[55], and is only slightly lower than methods designed with a focus on accuracy, such as SKFlow[36] and FlowFormer[15]. Compared to Flow1D[48], a notable memory-efficient method, our performance on the Sintel (Final) test set improved by 26%, and we led by 12% on the KITTI test set. Figure 1 and Figure 5 display some predictive results of our method and Flow1D on the KITTI and Sintel datasets. It is evident that our approach preserves more fine structure and performs better in large textureless regions. Figure 4 demonstrates that our method performs better in handling textureless regions even when compared to some accuracy-focused approaches.

We also conducted experiments on high-resolution images from the DAVIS dataset at both 1080P and 4K resolutions. As shown in Figure 2 and Figure 6, our HCVFlow exhibits superior performance in handling fine details, achieving accuracy noticeably better than Flow1D while consuming a similar amount of memory. Furthermore, we only need 6GB of memory to process 4K images, whereas RAFT fails to handle them on 48G A6000 GPU due to OOM.

### 5 Conclusion

RAFT and its successors achieve high accuracy in optical flow estimation using 4D global cost volumes. However, their memory consumption increases quadratically with image resolution, limiting their applicability to high-resolution images. In this paper,

**Table 6: Benchmark performance on Sintel and KITTI datasets.**

| Method | Sintel (test) | | KITTI test |
|---|---|---|---|
| | Clean | Final | |
| FlowNet2[18] | 4.16 | 5.74 | 11.48 |
| LiteFlowNet2[16] | 3.48 | 4.69 | 7.74 |
| PWC-Net+[35] | 3.45 | 4.60 | 7.72 |
| HD3[53] | 4.79 | 4.67 | 6.55 |
| IRR-PWC[17] | 3.84 | 4.58 | 7.65 |
| VCN[52] | 2.81 | 4.40 | 6.30 |
| DICL[38] | 2.12 | 3.44 | 6.31 |
| MaskFlowNet[55] | 2.52 | 4.17 | 6.10 |
| RAFT[37] | 1.61 | 2.86 | 5.10 |
| GMA[20] | 1.39 | 2.47 | 5.15 |
| SepaFlow[54] | 1.50 | 2.67 | 4.64 |
| GMFlow[49] | 1.74 | 2.90 | 9.32 |
| GMFlow+[50] | 1.03 | 2.37 | 4.49 |
| SKFlow[36] | 1.28 | 2.23 | 4.84 |
| FlowFormer[15] | 1.18 | 2.36 | 4.87 |
| DEQ-RAFT[1] | 1.82 | 3.23 | 4.98 |
| EMD-L[8] | 1.32 | 2.51 | 4.51 |
| SCV[21] | 1.72 | 3.60 | 6.17 |
| Flow1D[48] | 2.24 | 3.81 | 6.27 |
| HCVFlow (Ours) | 1.69 | 2.81 | 5.54 |

we propose a novel approach for constructing a hybrid cost volume (HCV) that achieves significantly lower memory consumption while maintaining high prediction accuracy. By decomposing the 4D cost volume into horizontal and vertical 3D cost volumes using the Top-k strategy and processing them with an aggregation module, we effectively reduce memory overhead while retaining the majority of matching information. Additionally, we construct a local 4D cost volume to supplement local information. By combining these cost volumes, our HCV not only drastically reduces memory usage but also achieves high prediction accuracy across various motion scenarios. We hope our research will advance the study and application of optical flow algorithms in high-resolution images and edge devices.

### Acknowledgement

This work is supported by the National Natural Science Foundation of China under Grant 623B2036.

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
