# OpenReview forum: "Hybrid Cost Volume for Memory-Efficient Optical Flow"
_acmmm.org/ACMMM/2024/Conference — MM2024 Poster_

### Official Review · Reviewer_uDBY · 2024-05-18

**Rating:** 5
**Confidence:** 2

**Summary:**

This paper proposed a memory-efficient optical flow method named HCV. The authors proposed a Top-K strategy that seperates the 4D cost volume into 3D, and it reduced memory consumption compared with the recurrent flow methods.

**Strengths:**

This paper is well-written and easy to follow, the main experimental result shows promising improvement.

**Limitations:**

It has not compared comprehensively with the recent advanced methods, e.g., FlowFormer++ (Shi et al., CVPR 23), 	DistractFlow (Jeong et al., CVPR 23)

**Suitability:**

3

---

### Official Review · Reviewer_3Erz · 2024-05-21

**Rating:** 2
**Confidence:** 3

**Summary:**

- This paper introduces an efficient Hybrid Cost Volume for estimating the optical flow.
- Different from previous flow methods introducing all-pairs cost volumes for high performance, the proposed HCVFlow proposes local 4D cost volume with a local search space, which reduces computational costs while retaining the performance.

**Strengths:**

- Employing local 4D cost volume scheme, reducing GPU memory usage by separating horizontal/vertical 3d global cost volume.
- Achieving comparable performance on multiple benchmark datasets with reduced computational costs.

**Limitations:**

- The crucial part of this paper is hard to follow. For instance, the process for generating cost volume, including calculating correlation and top-k strategy should be visualized with a description for better understanding. Like Figures in the Flow1D paper.
- For selecting horizontal/vertical Top-K pixels from the cost volumes, the spatial information of feature or cost volumes can be broken or no longer valid due to selection. Additional concerns about this problem need to be stated or properly described.
- Ablation study needs to be conducted more comprehensively. There are many hyper-parameters that can potentially impact performance in this work. Such as displacement radius D, number of K in Top-K, 4D local volume construction radius lr.
- In Tab. 1, each GCV and LCV consume 0.32G for 448 $\times$ 1024, why does the GCV+LCV consume only 0.38, which is much lower than the sum of both settings? For GCV and LCV, the network structure needs to be described more comprehensively.
- The visual results in Fig. 2,4,5, authors should display more candidate or competitive works to show superiority.
- Overall, the quality of the writing and the contribution are insufficient. There are lack of notations and details in the overview (Fig. 3), which hinders understanding. Further, compared to previous flow estimation methods, authors achieve marginal improvement on only memory consumption. For instance, in Tab. 3, the RAFT shows the best performance on most metrics, including the parameters with slightly larger memory usage.

**Suitability:**

2

---

### Official Review · Reviewer_WNGF · 2024-05-24

**Rating:** 2
**Confidence:** 3

**Summary:**

This work proposes a memory-efficient optical flow network, named HCVFlow. HCVflow adopts a novel Hybrid Cost Volume for memory-efficient optical flow, named HCV. HCV employs a Top-k strategy to construct 3D cost volumes, aggregates the initial 3D global cost volume, and constructs the 4D local cost volume. Experiments show that HCVFlow significantly reduces memory consumption while ensuring high accuracy.

**Strengths:**

1. The writing is clear.
2. The Hybird Cost Volume combines the strength of global cost volume and local cost volume, which is validated to be effective.

**Limitations:**

1. The main contribution of this work is the combination of global cost volume and local cost volume, which is limited. Moreover, in ablation study, the GCV+LCV retains the efficiency loss in trade-off accuracy improvement, which keeps a balance between efficiency and effectiveness. However, it is hard to conclude that it is the combination of these two cost volumes that introduces the efficiency improvement.

2. The improvement is limited as compared to accuracy-oriented methods. Compared to RAFT, the accuracy performance on KITTI-test and Sintel-clean and efficiency performance (memory usage/ inference time) are inferior.

3. The contents in Table 3 and Table 4 are the same.

**Suitability:**

3

---

### Official Review · Reviewer_hLNK · 2024-05-24

**Rating:** 4
**Confidence:** 3

**Summary:**

This paper presents a Hybrid Cost Volume (HCV) method to reduce memory consumption of optical flow estimation. Based on the observation that the global 4D cost volume used by RAFT has a spatial complexity growing quadratically with increasing image resolution, the proposed HCV mitigates the memory overhead by Top-k strategy. However, additional clarification along with experimental comparisons may be needed for better evaluations of the proposed Top-k strategy.

**Strengths:**

1. This paper focuses on optical flow estimation, which is a critical component of motion analysis for various multimedia applications.
2. The proposed HCV method is devoted to the memory efficiency of optical flow algorithms by improving the cost volume. The improvement is achieved by building 3D global cost volumes, aggregating information via a Top-k strategy and a separable aggregation module, and constructing local 4D cost volume.

**Limitations:**

1. Since Table 3 seems to be a replica of Table 4 and not be referred by the main content, the tables require an update.
2. Since the proposed HCVFlow architecture is trained through Adam that involves gradient descent methodologies, Section 3.1 only mentions the forward pass of TopK operation, missing the backward propagation regarding the TopK operation. Could the selection of the largest K values bring gradient loss?
3. Since the Top-k strategy is a key contribution to the improvement of memory-efficiency, it may lead to a question of how to choose k. Though Table 2 provides results of k=8, additional results of different k-values may better present how the choice of k affect the trade-off between memory-efficiency and noise reduction.
4. HCV approach is described to have competitive accuracy compared with RAFT but the Figures only demonstrate comparisons with Flow1D, thus qualitative comparisons regarding HCV approach against RAFT and other SOTA methods may be added.

**Suitability:**

3

---

### Meta-Review · Area_Chair_g8Hg · 2024-06-30

**Recommendation:** Accept (Poster)
**Confidence:** 5

**Metareview:**

This paper got one weak Accept, one Borderline Accept and two weak Reject. The studied problem is quite important for various multimedia applications, and it has been studied in previous works. The memory-efficient optical is achieved by building 3D global cost volumes, aggregating information via a Top-k strategy and a separable aggregation module, and constructing local 4D cost volume. The proposed idea is novel and the experimental results demonstrate its effectiveness. I recommend to accept to ACM MM. However, the authors should revise the paper according to the review comments.